



# GEOMAPLEARN 1.0: Detecting geological structures from geological maps with machine learning

David Oakley[1,2], Christelle Loiselet[1], Thierry Coowar[1,2], Vincent Labbe[1], and Jean-Paul Callot[2]

[a]Bureau de recherches géologiques et minières (BRGM), 3 Av. Claude-Guillemin, BP 36009, Orléans cedex 2, 45060, France

[2]LCFR, E2S UPPA, CNRS-TotalEnergies-UPPA, University of Pau and Adour countries (UPPA), Av. de l'université, BP 576, Pau, 64012, France

*Correspondence to*: David Oakley (David.Oakley@glasgow.ac.uk)

**Abstract.** The increasing availability of large geological datasets together with modern methods of data analysis facilitate a data science approach to geology in which inferences are drawn from geological data using automated methods based on statistics and machine learning. Such methods offer the potential for faster and less subjective interpretations of geological data than are possible from a human interpreter, but translating the understanding of a trained geologist to an algorithm is not straightforward. In this paper, we present automated workflows for detecting geological folds from map data using both unsupervised and supervised machine learning. For the unsupervised case, we use regular expression matching to identify map patterns suggestive of folds along lines crossing the map. We then use the hdbscan clustering algorithm to cluster these possible fold identifications into a smaller number of distinct folds, the number of which is not known a priori. For the supervised learning case, we use synthetic models of folds to train a convolutional neural network to identify folds using map and topographic data. We test both methods on synthetic and real datasets.

## 1 Introduction

The correct identification and interpretation of geological structures, such as folds and faults, is necessary in many applications including exploration for hydrocarbons, minerals, and groundwater, geological storage of wastes and energy resources, and seismic hazard analysis (Fossen, 2010; Roger, 2010; Brandes and Tanner, 2014; Bond, 2015; Li et al., 2019). On geological maps, structures can be identified manually based on an interpreter's knowledge of geological principles. Interpretation of geological data sets, including maps, is inherently subjective and uncertain (Bond, 2015), and it can be a time-consuming process. If the interpretation process can be automated, it could be made faster, more reproducible, and better able to handle large data volumes. With maps from many geological surveys now readily available in digital form, there is an opportunity to derive new insights from these old datasets using computerized analyses (e.g. Allmendinger, 2020). More broadly, there is a need to develop applications of machine learning for use throughout the geosciences (Bergen et al., 2019),





so automated structural interpretation can play a role within larger automated or semi-automated workflows. In particular, it is complementary to automated geological mapping methods (e.g. Cracknell and Reading, 2014), as it provides the next step of automatically interpreting the mapped structures. Further, three-dimensional geological modelling benefits from constraints based on knowledge of geological structure (Wellmann et al., 2014; Laurent et al., 2016; Grose et al., 2019), suggesting that automated structure identification could contribute to a larger automated geological model-building workflow.

35   Previous work on the automatic detection of geological structures on maps is limited. A greater body of work exists on the related problem of automatic classification of lithology from remote sensing and airborne geophysical data, as in the work of de Carvalho Carneiro et al. (2012), Cracknell and Reading (2013, 2014), Kuhn et al. (2018), Gillfeather-Clark and Smith (2018), and Bressan et al. (2020), as well as some work on fault and lineament detection from such datasets (e.g. Vasuki et al., 2014; Middleton et al., 2015; Aghaee et al., 2021). Another problem that has received greater attention is automatic

interpretation of seismic reflection data, including the identification of faults (e.g. Wu et al., 2019, 2022; Cunha et al., 2020; An et al., 2021, 2023; Gao et al., 2022; Wang et al., 2023) and salt structures (e.g. Shi et al., 2019. Muller et al., 2022). Identification of folds and other structures in magnetic data has also recently been investigated by Guo et al. (2021). In the interpretation of geological maps specifically, an early use of computers to extract new insights was the work of Ichoku et al. (1994), who developed a method for automated construction of geological cross sections from maps. More recently, Huang et

al. (2023) developed a method for automated cross section construction that includes identification of folds within the cross section. Of particular relevance to our work, Li et al. (2019) developed a method for identifying geological folds from vector geological maps using attributed relational graphs and formal grammar. While capable, this method assumes that the folds have an elliptical shape in map view and requires perfectly symmetrical folds with visible fold cores – requirements which natural folds will not always satisfy and that we seek to remove in this work. We, therefore, propose two alternative workflows

for detecting structures on a geological map, which we implement in the GEOMAPLEARN codes: one based on unsupervised clustering and the other using a convolutional neural network. Similarly to Li et al. (2019), we focus on detecting geological folds as a proof of concept, with the intention that our methods may be generalized to other geological structures in the future.

## 2 Methods

### 2.1 Data

55   The data needed for GEOMAPLEARN are a shapefile of geological unit polygons and a digital elevation model (DEM). The shapefile should include an attribute giving the relative age of each geologic unit, with 1 for the youngest unit. This relative age information is necessary for identifying the age relationships that are indicative of either anticlinal or synclinal folding. A relative age of 0 indicates that the program should ignore a unit, which we use to exclude Quaternary deposits from the analysis. We test GEOMAPLEARN on both synthetic and real-world data. For the real-world data, we use 1:50,000 scale

maps from the BRGM geological map database and a DEM from IGN data. All data processing and analysis are performed





using the Python programming language, with the *GeoPandas* (Jordahl et al., 2022) and *Shapely* (Gillies et al., 2023) libraries used to handle geospatial and geometric data.

## 2.2 Clustering-Based Unsupervised Learning Method

Our unsupervised learning method involves sampling a vector geological map using rays across the map, detecting patterns of interest, and clustering the detected features. Figure 1A summarizes the steps involved. We illustrate the process on a synthetic geological map (Fig. 2), with the results of the different steps shown in Fig. 3.

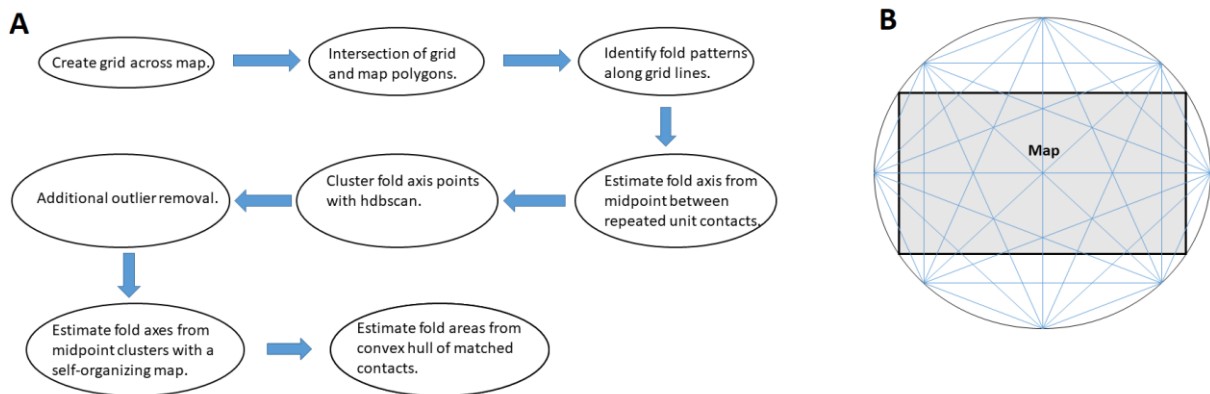

**Figure 1: (A) Flow chart of the unsupervised clustering-based fold detection algorithm. (B) Example showing how a grid of rays is created across a geologic map, from which to extract data.**


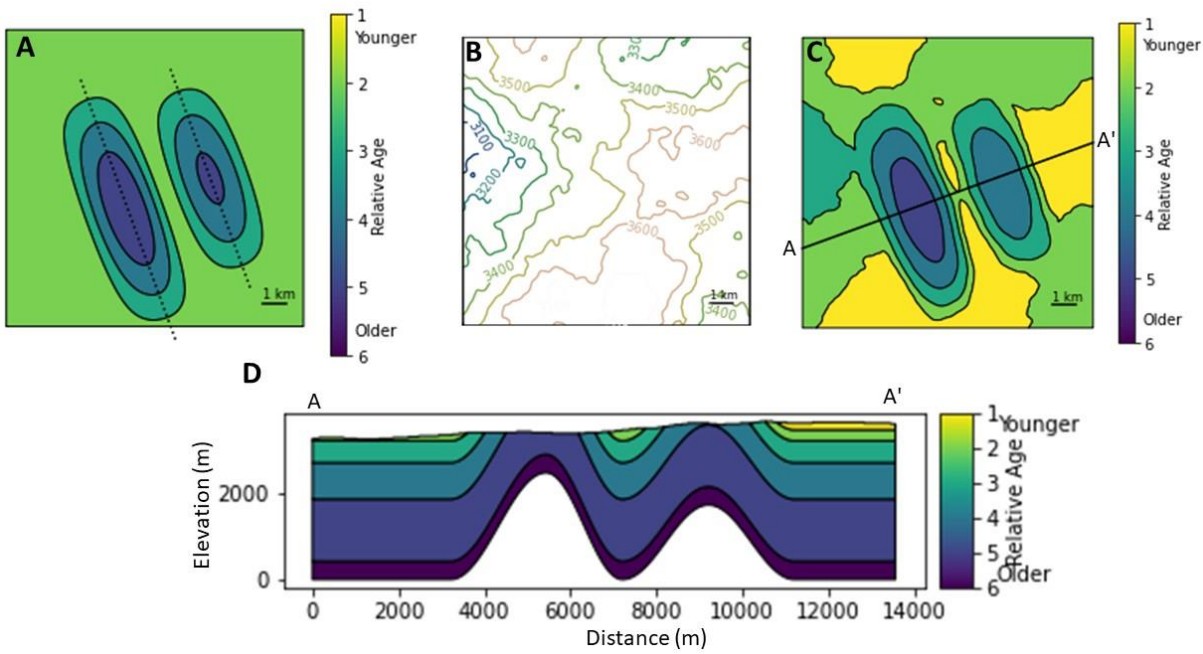





**Figure 2: Synthetic geological model used to illustrate the unsupervised learning method. (A) The geological map in the absence of topography (at approximately the mean elevation of B), with the axes of the two anticlines marked by dotted lines. (B) The topography of the model (with elevations in meters above the base of the model). (C) The geologic map formed by the intersection of the topography with the geologic model. (D) A cross section illustrating the two anticlines and the syncline in the model.**

The first step of our method is to extract one-dimensional, linear samples of the map to analyse for patterns indicative of folding. This approach is based on the initial work of Loiselet et al. (2016) who propose to sample a three-dimensional geological model along straight lines drawn in the model space. The sampling method provides an interface to the model using only two predicates related to (1) the geological domain that any point lies in, and (2), the geological contact (horizon or fault) that an arbitrary ray might intersect. Hence, answering only these two questions allows retrieving all the topological information automatically from the model and generating a model representation on demand (log, profiles, 3D gridding, etc.). Here we apply the same idea for sampling a two-dimensional geological map. To sample the map evenly without regard to the orientation of structures, we create a grid of sample lines within a circle surrounding the map (Fig. 1B). The circle is divided into equally spaced nodes, and lines are drawn connecting all possible pairs of nodes. (We use 80 nodes for the examples shown in this paper, but for better visual display, only 8 are shown in Fig. 1B.) We then find the intersection of each line with the map polygons, creating a series of smaller line segments within each line, with each segment corresponding to a geological unit (Fig. 3A). Thus, we identify all units that a line crosses and the order in which they are crossed.

Where a line crosses the edge of a polygon, we also estimate the strike and dip of the contact. This is done by taking the locations and elevations of vertices of the polygon within a given distance of the crossing point and solving for the best-fit plane using the method described by Allmendinger (2020), which is the many-point extension of the three-point problem. The distance used should be small enough to capture areas of high curvature but large enough to work in areas without large topographic variations and is likely to be map dependent. Here we use a distance of 250 m, which we found to work reasonably well for all the maps considered in this study.

Following Li et al. (2019), we take the key identifying feature of a fold on a geological map to be the repetition of geological units on either side of the fold core, creating a symmetrical pattern around the fold axis. If strata get progressively younger to either side of the core, then the fold is an anticline, and if they get progressively older, it is a syncline. Unlike Li et al. (2019) we do not require a specific core unit to be identified; we only require that the appropriate age pattern be identified on both limbs of the proposed fold. To search for symmetrical patterns of repeated units, we convert the pattern of geological units along each line into a string of text characters. Each geological unit is given a unique identifier, and the identifiers of all units crossed by a line are concatenated together in order. For example, the string 'ABCBA' would indicate that three unique geological units, 'A', 'B,' and 'C' were crossed by a line, with 'A' and 'B' being encountered twice and 'C' in the middle. Such a pattern could indicate the presence of a fold, since 'A' and 'B' are repeated symmetrically about the 'C' core. Since





105 Quaternary sediments may cut across bedrock structure, we exclude identifiers for Quaternary units in the text string, and we combine any occurrences of the same unit that are separated only by a Quaternary unit.

**Figure 3: Illustration of clustering-based unsupervised fold identification using the synthetic geological map from Figure 2. The relative unit ages are now represented in grayscale in order to show the fold identification process more clearly. (A) One of the rays used to sample the map divided into segments corresponding to the geologic units of the map. (B) All ray segments along which possible folds were identified. (C) The midpoints of the possible fold segments. (D) The midpoints after rejection based on bedding orientation to exclude fold-mimicking topography. (E) The midpoints as clustered by hdbscan. (F) The points to be used for finding**



the fold axes, which exclude points from E for which rays cross at a high angle to bedding strike. (G) The fold axes fit to the points in F. (H) The line segments corresponding to the midpoints in E. (I) The fold areas identified by finding the convex hulls of the line segments in H.

To detect fold-like symmetric patterns automatically, we make use of regular expressions (Erwig and Gopinath, 2012) to detect patterns indicative of folding in the text strings created from the data extracted along each ray. We use two expressions:

$$(\text{\textbackslash}w)(\text{\textbackslash}w)(\text{\textbackslash}w)*?\text{\textbackslash}2\text{\textbackslash}1 \qquad (1)$$

and

$$(\text{\textbackslash}w)(\text{\textbackslash}w)\text{\textbackslash}1. \qquad (2)$$

In these expressions, '\w' is used to match a character belonging to the 'word' character class (which includes all letters). The parentheses around '(\w)' define it as a group that can be referenced later by a number. The '\1' and '\2' terms reference these groups, meaning that the characters matched by the first and second '(\w)' groups must be repeated. The '*' sign means that there can be additional results between the two matched groups. Finally, '?' means that the shortest match that meets the criteria will be used. Therefore, expression (1) looks for places where two units are repeated symmetrically, with any number of intervening units allowed, and expression (2) looks for places where a single unit is repeated immediately on either side of a core unit. In both cases, a single geological contact (as defined by the two units that are in contact) is encountered twice by a ray, but the order in which the two units are intersected is reversed between the two occurrences, as is expected to occur when the ray crosses a fold.

The string of characters is analysed from left to right. If a symmetry is detected, its location and identity are saved. The starting point for the analysis is then moved to after the first occurrence of the matched unit(s), and the regular expression matching is performed again until no more symmetries are detected. This process is repeated for each ray in the grid of rays used to sample the map. The result is a group of ray segments, each of which crosses a possible fold (Fig. 3B).

For each ray segment, we estimate the location of the fold axis as the midpoint between the matched contacts (Fig. 3C). We further classify the detection as a possible syncline or anticline based on the age relationships of the matched units. If age increases towards the center, the fold is classified as an anticline. If age decreases towards the center, it is classified as a syncline.

As can be seen in Fig. 3C, the possible folds detected at this point can include fold-mimicking topographic patterns. When horizontal or homoclinal stratigraphy is intersected by topography, ridges will show younger units surrounded by older, as in synclines, and valleys will show older units surrounded by younger, as in anticlines, and these patterns will be matched by the regular expressions (1) and (2). Examples of this phenomenon occur toward the sides of the synthetic geological map in Figs. 2 and 3. To distinguish these cases from actual folds, we consider the orientation of bedding. Similarly to Li et al. (2019), we reject matches for which the dip of either limb is < 5°. We also require that the limbs of anticlines should dip away





from the fold core, while those of synclines should dip towards the fold core, and we reject matches that fail to meet these criteria. These requirements remove most topography-related patterns that were initially misidentified as folds (Fig. 3D).

Given the dense grid that we use, any fold on the map is likely to be crossed by many lines, each of which will produce an estimate of a point on the fold axis. These points can thus be expected to cluster around the true fold axis, while multiple
folds will produce multiple separated clusters. Many different machine learning clustering algorithms exist, including several that are implemented in the widely-used *scikit-learn* python library (Pedregosa et al., 2011). Most of these algorithms require the number of clusters to be specified, and we do not know the correct number of clusters (the number of folds) ahead of time. An algorithm that avoids this requirement is dbscan (Ester et al., 1996). The dbscan algorithm requires two user-specified parameters: *eps* and *min_samples*. Points within a distance *eps* from each other are considered to be neighbours. A point with
at least *min_samples* neighbors is considered to be a core point of a cluster. All core points that are neighbors of each other as well as all the neighbours of those core points are grouped together in a cluster. In this way, multiple clusters may form if multiple groups of core points exist that are not connected to each other. Points not within *eps* distance of a core point of any cluster are considered to be outliers. A weakness of the dbscan algorithm is that the *eps* parameter can be difficult to choose without some prior knowledge of the expected clustering. To reduce this need for user intervention, we use a variant of dbscan
called hdbscan, which attempts to determine the distance parameter automatically (Campello et al., 2013; 2015; McInnes et al., 2017). The only required user-chosen parameter is the minimum cluster size (*min_cluster_size*). (By default, *min_samples* is set equal to *min_cluster_size*, which we do not change, although in principle they can be different.), To avoid getting a lot of very small clusters, we also find it necessary to set the optional *cluster_selection_epsilon*, which specifies a minimum distance below which clusters will be merged together (Malzer and Baum, 2020). Setting a minimum distance between clusters
is less constraining than setting a specific value for *eps*, but it does introduce some additional user choice. Thus, hdbscan is somewhat more flexible than dbscan but still requires user-specified parameters that limit its ability to be used in a completely automated manner.

Using hdbscan, we cluster the estimated fold axis points (Fig. 3E). We perform the analysis separately for points classified as belonging to anticlines and synclines, since points from a different fold type should not belong to the same fold.
Since we are clustering the midpoints of ray segments connecting matched contacts on either side of a fold, there is a risk that some midpoints may cluster close together even when the endpoints of the segments are far apart and do not actually belong to the same fold. To reduce the risk of this occurring, we perform outlier removal on the segment end points using the LocalOutlierFactor algorithm from *scikit-learn* (Breunig et al., 2000). If either endpoint of a segment is deemed to be an outlier by this analysis, then the segment and its midpoint are removed from the cluster.

To identify the fold axis, we fit a line to the points in each cluster. As midpoints of segments crossing the fold, they are expected to cluster around the fold axis, albeit with some scatter. To remove rays that cross only the edge of a fold, we use only clustered points for which the angle between the ray and the strike of the matched contacts is greater than 60° (Fig. 3F). This restriction removes midpoints from rays that just slightly cross the edge of a fold. Using only these points, we fit a one-





dimensional self-organizing map (SOM) (Kohonen et al., 1982) to each cluster (Fig. 3G). We use a method described by
Gorzałczany and Rudzinski (2018) to grow the length of the SOM to fit the number of points to be clustered.

Finally, in addition to the fold axis, we identify the area affected by folding in each map. Since each clustered point is the midpoint of a line segment connecting two matched contacts, our clusters of points are also clusters of line segments (Fig. 3H). We find the convex hull of the endpoints of these line segments in order to estimate the area affected by folding (Fig. 3I).

### 2.3 Supervised learning

The clustering-based method described above makes use of unsupervised machine learning. An alternative is to use supervised machine learning, in which a neural network is trained to identify folds based on examples. Unlike the method described above, this method does not require the explicit definition of a set of rules to identify a fold or the specification of clustering parameters, but it does require a set of labelled training models for the neural network to learn from.

We implement supervised learning on raster versions of the geologic maps, which allows us to take advantage of
existing work on image recognition using raster images. We use the U-Net convolutional neural network architecture (Ronneberger, 2015), implemented with *TensorFlow* (Abadi et al., 2015) and *Keras* (Chollet et al., 2015). This architecture, originally developed for medical image segmentation, segments an image by classifying each pixel as belonging to one of a specified number of classes, corresponding to types of objects to be identified in the image. In the geosciences, it has been used previously in automated seismic interpretation applications (e.g. Muller et al., 2012; Wu et al., 2019; Wang et al., 2023).

The input to our neural network consists of two raster images: a geologic map, classified by the relative age of the units, and a digital elevation model. Both images are scaled to the range [0,1]. The elevation is randomly generated without any relationship to folding and is, therefore, not used to help identify folding. Rather, it is included so that the neural network can learn to interpret the map patterns produced by the intersection of stratigraphy and structure with topography.

We train the neural network on synthetic maps derived from randomly generated geological models. These models
are created by starting with an initially horizontal or sub-horizontal stratigraphy, applying uplift to simulate folding, and finding the intersection of a topographic surface with the three-dimensional model. Synthetic fold geometries are created with a simple mathematical function for a sinusoidal fold, with randomly chosen values to define amplitude, wavelength, asymmetry, and orientation. The deformation consists purely of vertical uplift, which simplifies the process of creating and labelling the synthetic geological maps. We also add some simple synthetic faults, again with purely vertical motion, with maximum
displacement proportional to fault length (Cowie and Scholz, 1992) and a displacement field that has an elliptical shape (Walsh and Watterson, 1987; Georgsen et al., 2012; Wu et al., 2020) along strike and decreases away from the fault (Cardozo et al., 2008). The fault model, which is a simplification of that used by Wu et al. (2020) is not meant to generate truly realistic faults, since fault identification is not our focus, but to teach the neural network to handle maps that include abrupt offsets due to faulting. Additional Gaussian noise is also added to the uplift to make the trained model resilient to data that do not perfectly
follow our simple folding model. Synthetic topography is generated randomly using Perlin noise (Perlin, 1985). Perlin noise



is a method often used for procedural terrain generation in computer graphics (Rose and Bakaoukas, 2016), which produces sufficiently realistic-looking topography for our purposes and has the advantage that different topographies can be easily generated by randomly changing a single seed number. Because real-world maps typically contain fluvial and other Quaternary deposits that cut across bedrock geology, random Quaternary deposits are created with a second Perlin noise field. Values of

that field below a cut-off value are assigned to the Quaternary and given a relative age code of 0. The model is trained only to identify anticlines, not synclines. This is done due to the fact that the model only explicitly creates anticlines, with synclines merely being the space between them, as well as the fact that the limb of an anticline is also the limb of the adjacent syncline, and the U-Net architecture assumes that each pixel fits into only one class. More details of the synthetic modelling process and the ranges of values used for it are given in Appendix A and Table A1.

For training, we create a dataset of 100,000 random synthetic models. The use of such a large ensemble helps to avoid over-fitting that can occur when the neural network is repeatedly trained on the same models, and the use of synthetic models allows a large training dataset to be generated rapidly. 15% of the synthetic models have no folds in them but have topography that can create fold-mimicking map patterns; they thus help the neural network to learn to distinguish these patterns from actual folds. The neural network is trained for 250 epochs of 800 models each, using the synthetic dataset. Another 200 synthetic

models are used as a validation dataset. The sparse categorical cross-entropy loss function in *Keras* is used to determine the misfit between models and neural network predictions. The decrease in this loss function over the 250 epochs is shown in Fig. 4. The outputs of the neural network are three raster images, of equal size to the input images, with values between 0 and 1. These give the probability that each pixel belongs to one of three classes: not part of an anticline (background class), within the area of an anticline, and along the axis of an anticline. The inclusion of a background class is necessary because U-NET

must classify every pixel as belonging to some class. The training process relies on randomness – in the synthetic models, the order in which the training models are used, and the initial values of the neural network weights – which introduces some, currently unquantified, uncertainty in the training process, but experiments with training multiple times showed reasonably similar results.

The process of classification with U-Net cannot be as readily illustrated as the clustering process in Fig. 3, since the

internal workings of the neural network are opaque. Nonetheless, in Fig. 5 we use the same two-anticline example to illustrate the input raster images (Fig. 5A-B) and output classifications in terms of probability (Fig. 5C-E), which compare well with the truth (Fig. 5F-H).





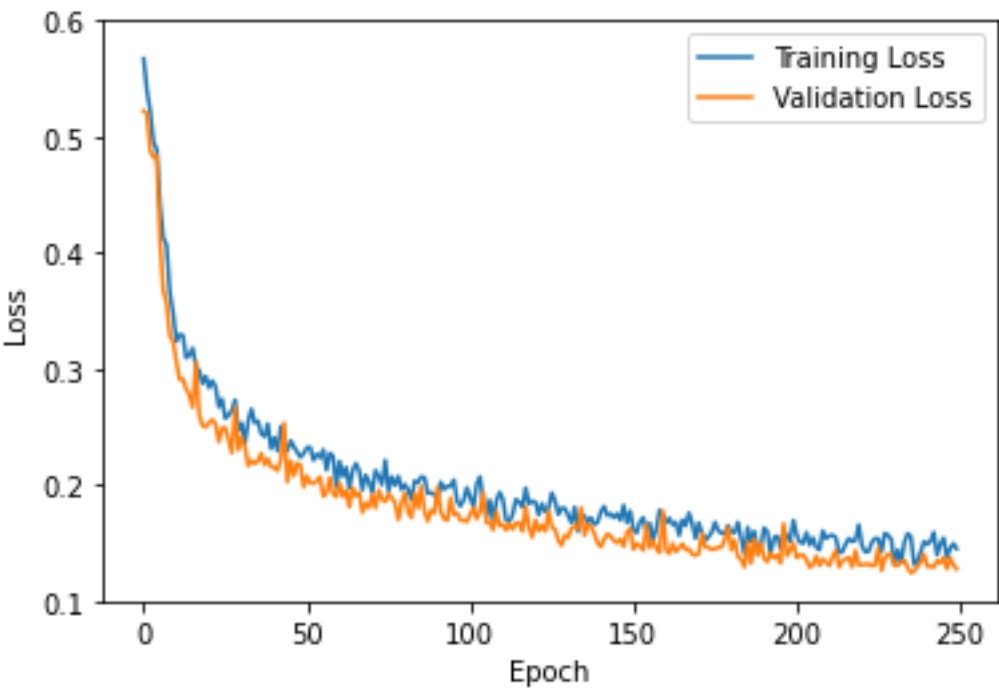

**Figure 4: Sparse categorical cross-entropy loss from training the convolutional neural network. Each epoch consists of 100 training**
**batches and 25 validation batches, and each batch consists of 8 synthetic maps.**



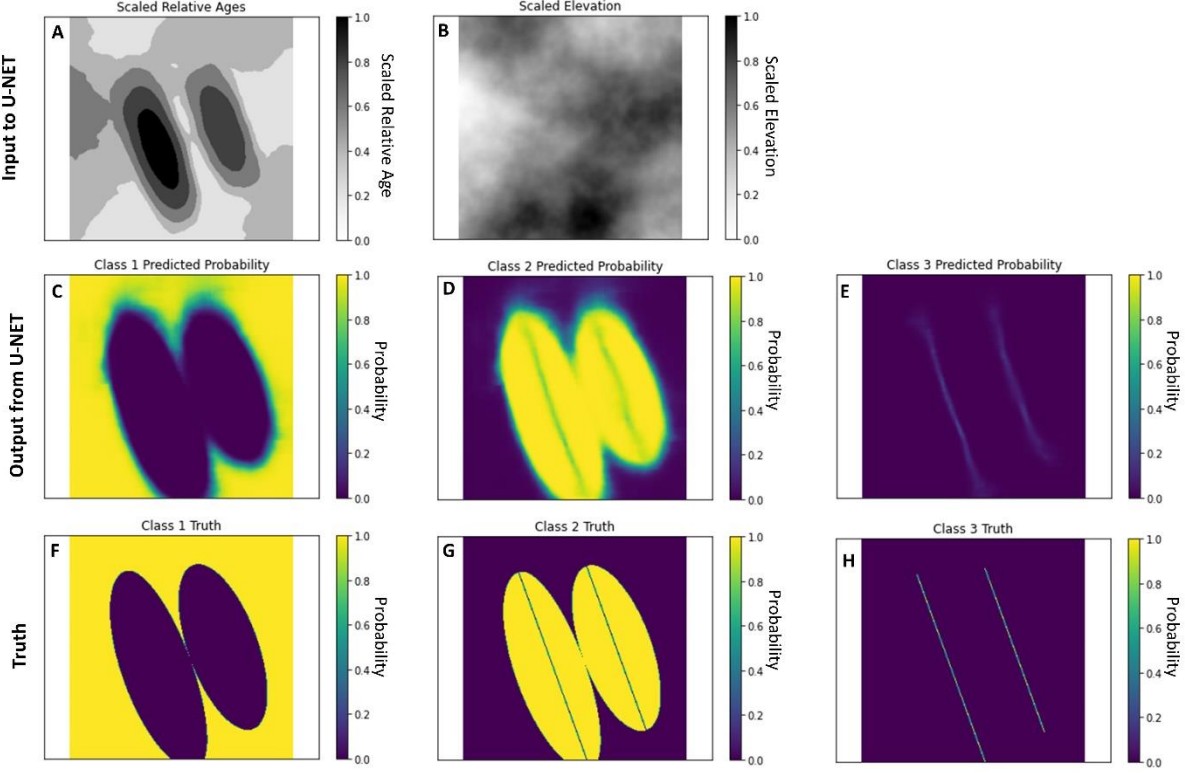

**Figure 5: The fold identification process with the convolutional neural network, illustrated using the example from Figures 2 and 3. (A and B) The input consists of two raster images, both scaled between 0 and 1, with the first giving the geological map in terms of the relative age of the units and the second giving the elevation. (C-E) The output consists of the probabilities that each pixel belongs to each of three classes: Class 1 for the background, Class 2 for off-axis parts of the fold, and Class 3 for the fold axis. The three class probabilities sum to 1 at each pixel. (F-H) The truth is plotted with each pixel given a probability of 1 for the correct class and 0 for the other classes.**

## 3. Results

### 3.1 Synthetic Maps

To validate the proposed methods, we first test them on a series of synthetic geological maps (Fig. 6), in addition to the synthetic map in Fig. 2. We consider five structural models, which illustrate main aspects of fold structure that we wish to capture: 1) A flat model with no folding, 2) A single symmetric, double-plunging anticline, 3) An asymmetric anticline, 4) An anticline with a sigmoidal shape, and 5) A map with five, interacting anticlines. We generate synthetic topography in two ways. In the first case, we use *Perlin noise*, as in the synthetic models for training the neural network, to produce moderately

realistic-looking terrain. In the second, we use a deliberately unrealistic "egg-box" shape formed by the sum of two perpendicular sine functions, which is chosen for its complexity and the fact that its interaction with a horizontal stratigraphy



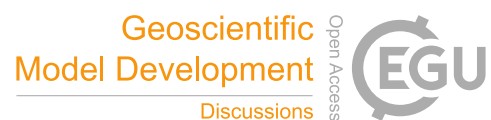

will produce many fold-mimicking map patterns. We apply the *Perlin noise* topography to all models and the egg-box topography to the flat, symmetric fold, and multiple folds models.

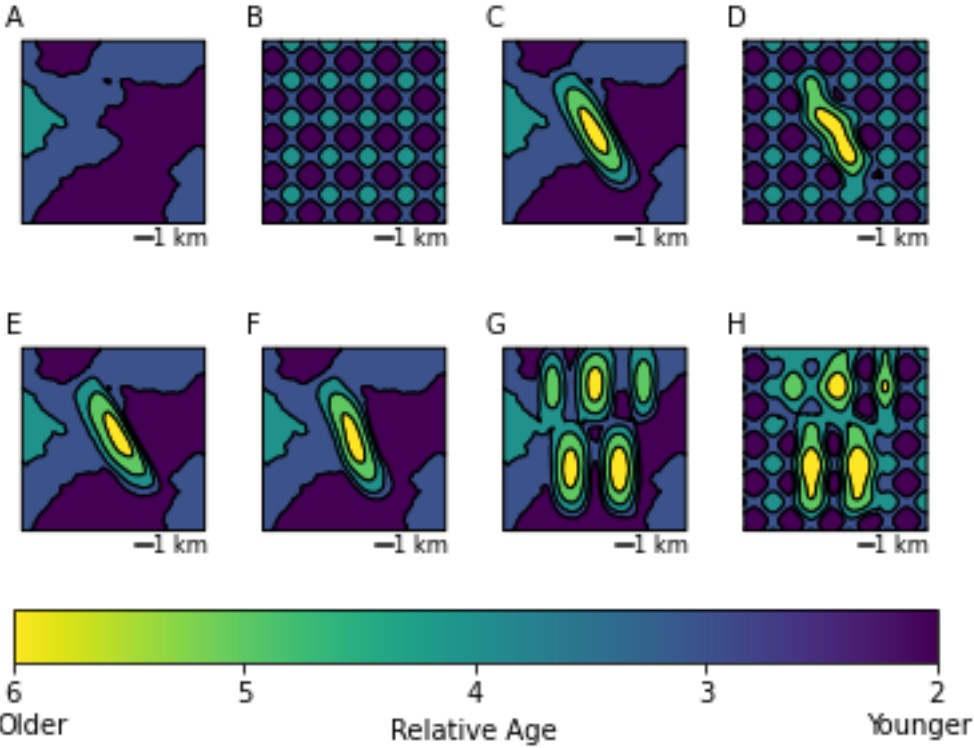

**Figure 6: Geologic maps of synthetic models colored by the relative age of the units: A) Flat model, B) Flat model with egg-box topography, C) Single symmetric anticline model, D) Single symmetric anticline model with egg-box topography, E) Asymmetric anticline, F) Sigmoidal anticline, G) Multiple folds, H) Multiple folds with egg-box topography.**

Figures 7 and 8 show the results of the unsupervised learning method on these examples. For *hdbscan*, we use
parameter values of *min_cluster_size* = 15 and *cluster_selection_epsilon* = 500. For the flat model, we successfully determine that there are no folds in the map, despite the presence of topography that creates fold-like patterns (Figs. 7A and 8A). This holds true even in the egg-box topography case, which presents many such patterns (Figs. 7B and 8B). (Note that this is highly dependent on the use of bedding orientation information in identifying the folds; if only the age relationships of geological units are used, then folds are incorrectly identified in this case, similar to what is seen in Fig. 3C.) For the single, symmetrical
anticline, a single fold is successfully identified with an approximately linear axis, although the axis does not extend quite to the plunging end of the fold (Fig. 7C). The fold area accurately covers most of the fold, although a small part of it is cutoff at the northwest end (Fig. 8C). With egg-box topography the fold is still successfully identified, although the interaction of the fold with this unusual topography creates a wavy shape to the fold that is not truly accurate (Figs. 7D and 8D). For the asymmetric anticline, the approximately correct fold area is still identified (Fig. 8E), but the axis is still in the center of the





fold rather than in its asymmetric position to one side (Fig. 7E) due to the limitations of using the midpoints of the lines as an estimate for the fold axis. For the sigmoidal anticline, the curved shape of the fold axis is successfully identified (Figs. 7F and 8F). In the multiple folds example, fold axes were identified on all of the anticlines and the synclines between them (Fig. 7G-H), although two syncline axes in Fig. 7 are joined through the saddle area between the ends of two other anticlines. The fold area polygons are also mostly reasonable, if a bit angular (Fig. 8G-H), although the polygon for the southwestern anticline in

Fig. 8G extends across the northwestern one as well. In summary, the method successfully identifies the existence of all folds in the models and approximately, but imperfectly, represents their geometry.

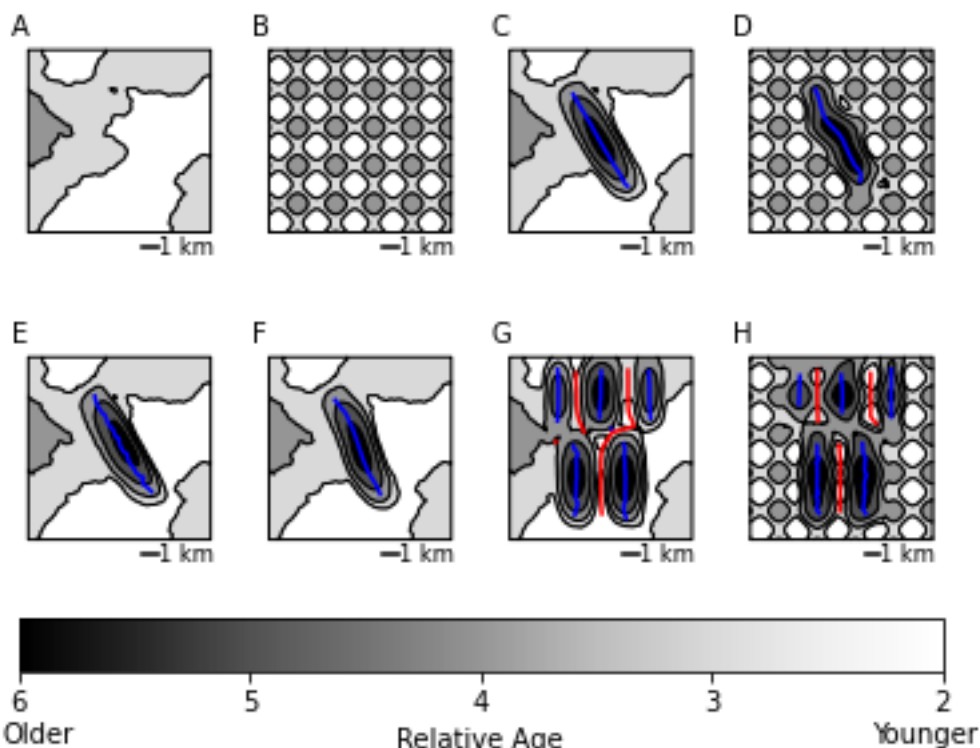

**Figure 7: Axes of folds identified by the unsupervised clustering method for the synthetic models in Figure 6. Axes are found by fitting a linear SOM to the midpoints of line segments connecting matched contacts on either side of the fold. Anticline axes are blue,**
**and syncline axes are red.**

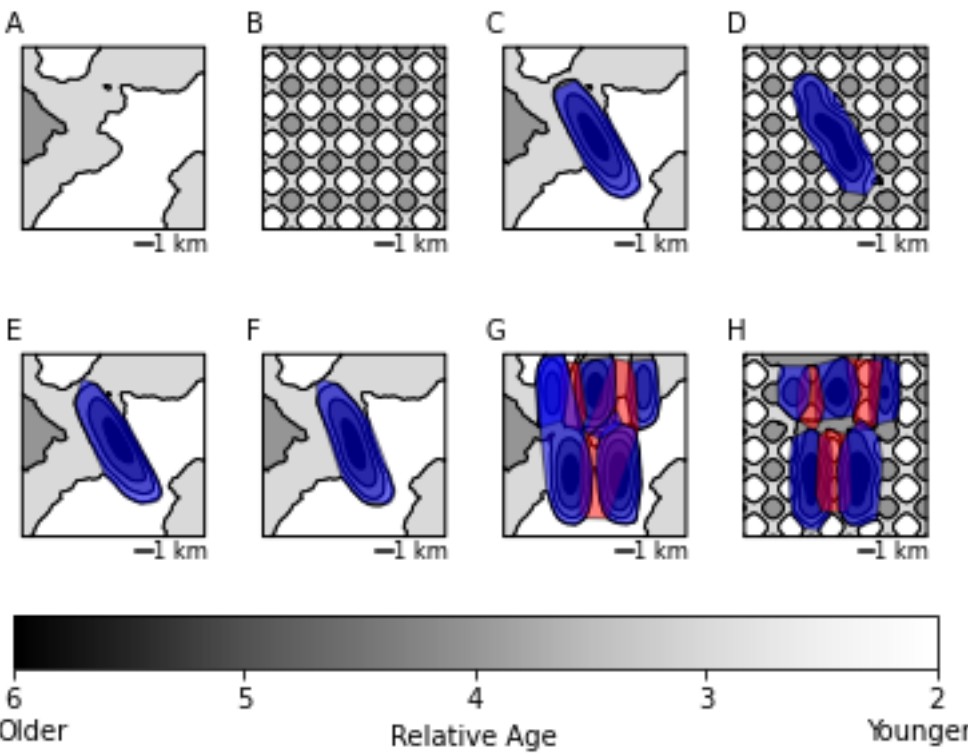

**Figure 8: Fold areas identified by the unsupervised clustering method for the synthetic models in Figure 6. Areas are the convex hulls of the line segments in each cluster. Anticlines are blue, and synclines are red.**

*U-Net* classification of these same models is shown in Figs. 9 and 10. Figure 9A-H shows the probabilities that each pixel of the map is intersected by the fold axis. While the maximum probabilities are relatively low (<0.5), the axes still show up as distinctly higher probability areas compared to their surroundings. Figure10A-H shows the probability that each pixel is part of the fold, which is calculated as the sum of the fold axis and off-axis fold area probabilities. These are identified with high certainty (probabilities near 1) and approximately match the true fold shapes as shown by the contours in Fig. 10. The

flat models show no folds (Fig. 9A-B, 10A-B), suggesting that the trained neural network has learned to distinguish topography-related map patterns from real folds. The single, symmetric fold model shows that it can correctly identify the fold axis (Fig. 9C-D) and area (Fig. 10C-D) without being fooled by topography, even in the egg-box topography case, although in that case it is slightly curved. For the asymmetric fold, (Fig. 9E, 10E) the detected axis closely follows the true offset axis, and for the sigmoidal fold (Fig. 9F, 10F), it has the expected sigmoidal curved shape, showing that the neural network can identify

these cases. For the multiple folds case, all anticlines and their axes are detected with the regular topography (Figs. 9G, 10G) and egg-box topography (Figs. 9H, 10H).





**Figure 9: Fold (anticline) axis detection by the convolutional neural network supervised learning method for the synthetic maps in Figure 6. Colors indicate the probability of each pixel being on the fold axis. Black lines show the true fold axes. Note that the color scale is different here than for the fold areas in Figure 10, due to the maximum probabilities of the fold axes being lower.**







**Fold Area Probability**

**Figure 10: Fold area detection by the convolutional neural network supervised learning method for the synthetic maps in Figure 6.**
**Colors indicate the probability of each pixel being within the area of the fold (including the axis).**

## 3.2 Real-World Maps

We test our methods on two real-world cases taken from BRGM 1:50,000 maps of France (Fig. 11). The first of these is the NNW-plunging Dreuilhe or Lavelanet anticline in the Ariège department of France, as shown on the 1076 Lavelanet map (Souquet et al., 1984). The anticline is a major structure of the sub-Pyrenean zone immediately north of the North Pyrenean Frontal Thrust (Grool et al., 2018). To avoid the more intensely faulted region in the hanging wall of the thrust, we interpret





only the northern part of the Souquet et al. (1984) map. Our second real-world case is the 0222 Esternay map of Weecksteen et al. (1968), which covers an area of the Brie Plateau in the Marne department. The region is part of the sedimentary Paris Basin, and erosion has formed ridges and river valleys that produce map patterns in which units are progressively older or

younger away from these topographic extremes. This map is, therefore, chosen to test the abilities of our methods to distinguish these topography-related features from actual folds.

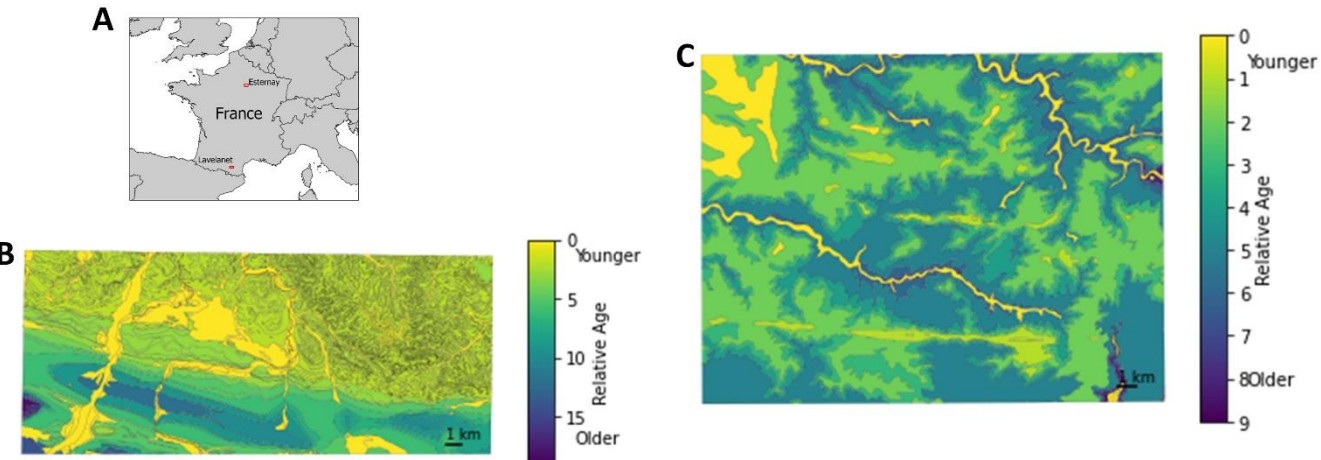

**Figure 11: Real-world maps from two sites in France (locations marked in A): (B) the Lavelanet anticline and (C) the Esternay**
**region. Map polygons are taken from the digital versions of the BRGM 1:50,000 maps 1076 (Lavelanet) and 0222 (Esternay), but they have been colored according to the relative ages of the units, which is the information used in our proposed methods.**

For the unsupervised clustering method, we use *hdbscan* parameters of *min_cluster_size* = 30 and *cluster_selection_epsilon* = 1000 m. The Lavelanet anticline axis is detected (Fig. 12), and a second anticline to the east is also
identified. Since there is continuous uplift between them, the two parts could also be considered part of the same structure, but the break coincides with a clear structural low and is not unreasonable. In addition, the syncline adjacent to the Lavelanet anticline and the plunging end of another anticline on the west side of the map are also identified. The fold areas, while reasonable in their positioning, have overly angular shapes compared to the shape of the actual folds on the map. For Esternay, our program successfully ignores the topography-related fold-like patterns and determines that the map contains no folds.





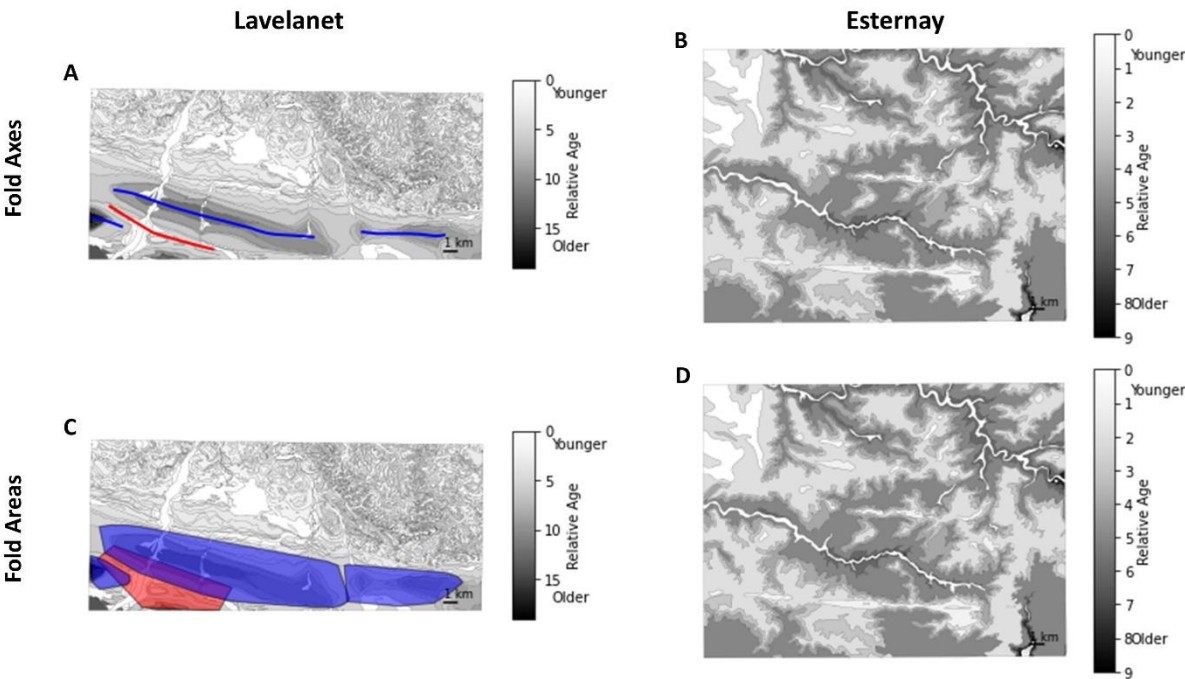


**Figure 12: Results of the clustering-based unsupervised learning method for the Lavelanet anticline and Esternay maps, showing the axes and areas of folds identified by the method. Anticlines are shown in blue and synclines in red. For Esternay, no folds were identified.**

The results of clustering with *hdbscan* are dependent on the *min_cluster_size* and *cluster_selection_epsilon* parameters. In Fig. 13, we experiment with changing these values. In Fig. 13A and C, *cluster_selection_epsilon* is reduced to 500 m. The anticline to the east of the main Lavelanet anticline has now been split into two parts, and the syncline is no longer detected. In Fig. 13B and D, *min_cluster_size* is reduced from 50 to 25. In this case, the Lavelanet anticline and the anticline to the east are merged. Further, a small anticline and a syncline spanning a large area are identified in the east and northeast,

largely in an area of fluvially disected homoclinal stratigraphy, which appear to involve topography-related patterns misidentified as folds.



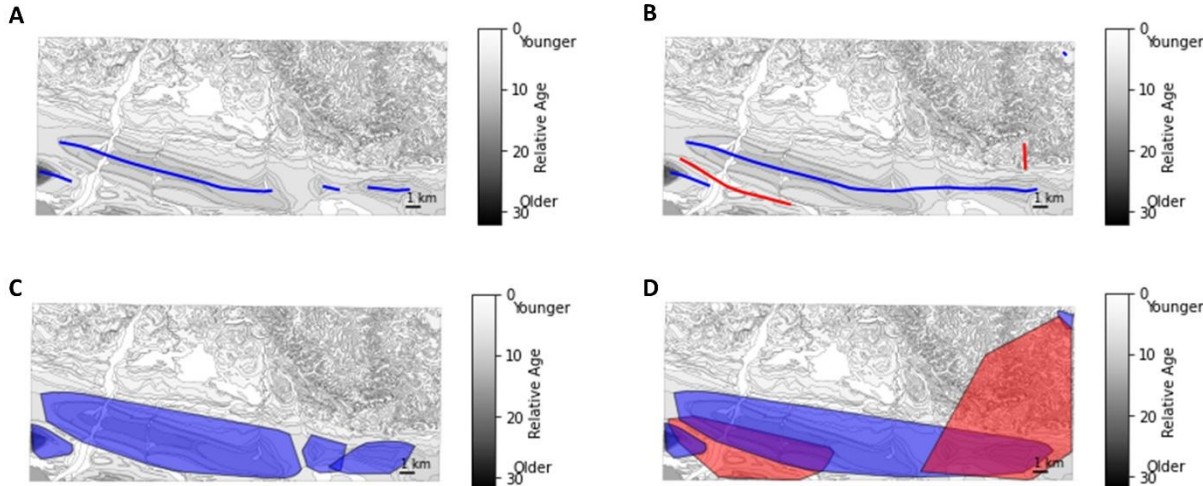

**Figure 13: Fold axes (A-B) and areas (C-D) of the clustering-based method for the Lavelanet anticline using different *hdbscan* clustering parameters than those in Figure 12. (A and C) The *cluster_selection_epsilon* parameter is reduced from 1000 m to 500 m. (B and D) The *min_cluster_size* parameter is reduced from 30 to 15. Compare to Figure 13, left column.**

The real-world maps are larger than the *U-Net* training maps, but we are able to interpret them with *U-Net* using a sliding window, similarly to Ronneberger et al. (2015), and results are shown in Fig. 14. The vector geological map, classified by relative age of the units as in Fig. 11B and C, is converted to a raster using ArcGIS. The raster contains a single channel giving the relative age, which is rescaled to the interval [0,1] (Fig. 14A-B). This along with a raster DEM, also rescaled to [0,1], is used as the input to the trained neural network. In the Lavelanet map, the neural network is able to detect the main Lavelanet anticline and its axis, as well as the additional anticline to the east and the plunging end of an anticline on the west side of the map. These results show that what was learned from synthetic models can be extrapolated to real-world structures as well. In the Esternay map, the neural network gives low to moderate fold probabilities over most of the map, but it incorrectly identifies a region on the west side of the map as containing folds, which appear to approximately follow river valleys. Thus, despite the ability of the neural network to separate true folds from topography-related patterns in synthetic cases (Fig. 9A-B, 10A-B) and in some parts of the Esternay map, this aspect of map interpretation can still pose a challenge in some real-world cases.





**Figure 14: Results of the convolutional neural network supervised learning method for the Lavelanet anticline and Esternay maps, showing the axes and areas of folds identified by the method. The fold area probability is the sum of the off-axis fold area probability (class 2) and the fold axis probability (class 3).**



## 4 Discussion

### 4.1 Unsupervised and Supervised Learning

In this paper, we have tested two machine learning approaches to the automatic identification of folds on geological maps: one based on unsupervised learning and one on supervised learning. Both have proven capable of identifying folds and fold axes on synthetic and real maps, although each has some advantages and disadvantages.

The unsupervised learning approach makes use of explicit rules for identifying folds, while the supervised learning method makes use of training examples from which rules are learned. In the absence of sufficient labelled training examples,

this difference favors the unsupervised approach, but we have largely been able to overcome this limitation by using synthetic data for training. Another advantage to the use of explicit rules in the unsupervised approach is that they can be readily understood by a human user, while the decision-making process of a trained neural network is opaque and can be evaluated only on the correctness of its results, not of its methods. Whether rules are explicitly defined or learned, they may not account for all possible variations of real-world geology, and such limitations can be identified and understood more easily in a set of

explicit rules than in a neural network. On the other hand, a neural network can represent a much more complex set of rules than can reasonably be defined manually. Finally, since the rules for identifying an anticline and a syncline are essentially opposite of each other, we can easily define both rules in the same manner in the unsupervised learning approach, while the supervised learning approach cannot readily make an analogy between them and is, at present, only able to detect anticlines.

A primary goal of our work is to reduce or eliminate the need for human involvement in the structure-detection

workflow. This goal is better achieved by the supervised learning method. The unsupervised learning workflow shows sensitivity of the results to the user-specified settings of the *dbscan* and *hdbscan* clustering algorithms (Fig. 13), which limits use of the method in a fully automated manner and requires manual tuning to any particular map under study. With the supervised learning method, in contrast, the neural network only had to be trained once and could then be applied to multiple different cases, as demonstrated with synthetic and real-world maps. However, both methods would likely require some human

review of the results to ensure that they are reasonable.

There are also differences between the two methods in terms of the form of their input data and output results. The unsupervised learning method works with vector data – the usual form of GIS geological maps. The supervised learning approach requires raster data, which may be less precise. For the real maps we use as examples, we converted the vector map to a raster with a 50 m cell size (the same as for the DEM we were using). If geological contacts are located with better than

50 m precision, then this conversion will entail some loss of data precision. However, this precision should be sufficient for the primary task of identifying map-scale folds, which are much larger than 50 m. Finally, the unsupervised learning approach explicitly separates the individual folds identified, while the supervised approach that we have used produces only a single mask of where folds are or are not present. This form of output is known as "semantic segmentation," while additional separation of the individual folds would be "instance segmentation" (Géron, 2023). A more advanced machine learning

architecture might be able to solve the instance segmentation problem in this case, but our work solves at least the major



problem of identifying which portions of the map are folded and is similar to U-NET-based seismic interpretation methods, which also focus on semantic segmentation (e.g. Wu et al., 2019, 2020; Wang et al., 2023). Overall, the use of raster datasets does add some limitations to the supervised learning approach as opposed to the unsupervised approach, but still provides meaningful and useful results.

## 4.2 Use of Synthetic Training Models

The use of synthetic data is crucial to the success of our supervised learning method. Without it, we would have had to assemble a large dataset of labelled real-world geological maps of folds for training. With synthetic models, we can rapidly produce many more datasets than are likely to be available from real-world data. Further, we can be sure that the fold extents and fold axis locations defined for the training datasets are exactly correct, while for real-world training data, these interpretations would be subject to the uncertainties of manual identification and conceptual biases introduced by the background of a particular interpreter (Bond et al., 2007). We also know exactly what geological processes are or are not simulated in a synthetic dataset and therefore what kinds of geological settings a neural network trained on that dataset may be suited to interpreting. Similar considerations to these have led to the use of synthetic geological models in training a neural network for identification of faults in seismic data by Wu et al. (2020) and Wang et al. (2023), of structures in magnetic data by Guo et al. (2021), and in developing a large dataset of 3D models for machine learning applications by Jessell et al. (2022).

The primary drawback of our use of synthetic models is that they cover only a limited range of geological models. Structures in our models are limited to a simple model of folding and an even simpler model of faulting. Stratigraphy is always conformable, and topography and the distribution of Quaternary sediments follow a simple noise model. In addition, all model parameters lie within a limited range (Table S1). The simple models may be insufficient for the neural network to be able to identify some real-world structures, and the training models may introduce their own biases, even as the bias of a human interpreter is removed. The misidentification of folds in the Esternay map, for instance, may indicate that the model is overly biased towards seeing folds. Nonetheless, we have shown with the Lavelanet map that our neural network trained on synthetic fold models can successfully identify real-world folds. The neural network also shows the ability to generalize beyond the exact forms of its training dataset. For instance, the folds in the training dataset all had perfectly linear fold axes, but when confronted with a test example having a sigmoidal fold axis (Figs. 9F and 10F), the neural network was able to correctly identify this axis shape.

## 4.3 Effects of Topography

With the focus of our structure detection work on folds, a major challenge has been distinguishing actual folds from fold-like patterns of geological units produced by the intersection of topography with a homoclinal stratigraphy. In both the unsupervised and supervised learning approaches, our solution has been to include topographic information along with the geological map data. Topographic data are widely available, so this requirement is unlikely to be a major limitation of either



method. As with the geological unit data, this data is used to define specific rules (by way of bedding orientations calculated from multi-point problems) in the unsupervised case and to train the neural network to learn its own rules in the supervised case. In the unsupervised learning case, Fig. 3C shows that topography-related fold-mimicking patterns will be initially

identified as possible folds by regular expression matching and thus the use of the topographic information and bedding orientation constraints is crucial (Fig. 3D) and is ultimately successful in separating these patterns from actual folds in several different example cases (Figs. 3, 7, 8, and 12). How the neural network uses topographic information internally is not clear, but Figs. 9A-B and 10A-B illustrate that it can learn to distinguish topography-related patterns from real folds when trained on synthetic models including such patterns. However, as seen in the Esternay case study (Fig. 14), the ability of the neural

network to generalize these principles to real-world maps is imperfect, perhaps due to differences between the synthetic and real topography. Improved interpretation of topography is, therefore, an area for future development that could significantly improve the supervised learning method.

The rules that the unsupervised learning method uses to remove initial fold identifications on the basis of bedding orientation may prevent the identification of some true folds. The requirement for dips greater than 5° could exclude some

extremely gentle folds, while the use of bedding dip directions in identifying synclines and anticlines will fail to identify folds with overturned limbs as well as some multiply folded structures (synformal anticlines and antiformal synclines). While we cannot determine if the neural network's internal rules prevent it from identifying these structures, we have only trained it on upright folds, so its application to other cases should be made with caution. Other types of folds are thus another area for future work.

So far, we have treated topography only as a hindrance to structure identification, and in our synthetic models, there is no correlation between topography and structure. In the real world, however, topography could be related to structure either through the different resistance to erosion of folded units or through the effects of uplift on landscape evolution in active tectonic settings. While consideration of these factors would add considerable complexity to the synthetic models, they could be considered as a possible source of information in the future.

**4.4 Extension to Additional Kinds of Structures**

In this work, we have focused on identification of folds and have used as examples maps in which folds are the principal structures to be observed. Many geological settings are, however, considerably more structurally complex, and our methods for the automatic identification of folds may serve as a starting point for the identification of geological structures more broadly. Both methods could in principle be extended to additional structures. Unsupervised learning requires the explicit

definition of a set of rules, which is likely to grow highly complex as the variety of structures to be considered increases, although it may be possible with the use of a geological ontology such as that of Brodaric and Cox (2020). For example, our rules for finding repeated units on either side of a fold do not currently account for faults, which may produce similar repetitions without a fold (although the use of bedding orientation will help to rule these out) or may break up the patterns indicative of



folding. To include fault identification in the method, it would therefore be necessary not only to define rules for identifying
faults, but also to define rules for interactions between faults and folds. With the supervised learning method, in contrast, it
would only be necessary to create training models involving both faults and folds, which is quite doable with relatively simple
geological modelling methods (e.g. Jessell et al., 2022), to add an additional output class to the neural network for identifying
them, and to retrain the neural network with the new training data. With the addition of further complications (e.g.
unconformities, intrusions, salt domes, etc.), defining explicit rules for all cases and their interactions would grow even more
difficult. A supervised, neural-network-based method is, therefore, probably the simplest option for generalization of the
methods described here to a wider range of geological structures, although both methods could in principle be generalized. A
convolutional neural network could also be trained to take additional image channels as input, which could allow for the joint
interpretation of map-view geophysical data (as in Guo et al., 2021) with geological maps and digital elevation models.

## 5 Conclusions

475        We have shown how both unsupervised and supervised machine learning methods can be used to detect folds in
geological maps, thus automating a process that traditionally relies on a human interpreter with conceptual understanding of
geological principles. Both methods prove reasonably capable, although challenges remain. The unsupervised, clustering based
method requires intelligent choices of clustering parameters by the user and in some cases produces overly angular or overly
large estimates of the area of a fold, even when the axis is better located. The supervised learning method is more fully
automated but can still have trouble in some cases, in particular in misidentifying some topography-related patterns as folds
as on the Esternay map.

        A particular issue that we have investigated is the ability of machine learning to deal with topography in map
interpretation. Topography significantly complicates the problem, and in both methods we have developed it has been
necessary to include topographic information along with the geologic map itself and to ensure that the machine learning method
makes use of it either through explicitly defined rules (in the unsupervised case) or through examples (in the supervised case).

        Synthetic models were key to our use of supervised machine learning in this application, since without them it would
have been difficult to compile the large and diverse training dataset needed for these methods. Our results have shown a
reasonable ability of the trained neural network to extrapolate from synthetic models to real-world data. However, in regard to
the relationship between map patterns and topography, the trained model remains better at interpreting synthetic models than
real world cases.

        While we have focused our work on folds, we aim to develop methods that can eventually be applied to the
identification of a wide range of geological structures. For the unsupervised learning method, this could begin with the same
method of map sampling by rays that we have used here but incorporate rules for identifying a wider variety of structures. For
the supervised learning method, the primary future need is to build synthetic models that incorporate a greater variety of labeled
structures.





## Appendix A: Description of the Process for Generating Synthetic Models

This appendix describes the process used for creating random geological models and resulting maps to train the convolutional neural network. The randomly chosen parameters and the probability distributions from which they are drawn are listed in Table A1. The same process, without the random elements, is also used to make the synthetic models in Figs. 2

and 6.

Models are built in a Cartesian ($x$,$y$,$z$) coordinate system, where $x$ is east, $y$ is north, and $z$ is up. The origin of each model is set at the southwest corner in $x$ and $y$ and at the base of the stratigraphic column in $z$. We train the convolutional neural network on 256 x 256-pixel raster images. In model coordinates, each pixel is given a size of 50 m, which is the same resolution as the digital elevation models we use with the example real-world maps. This results in a model grid of dimension

12,800 m in the $x$ and $y$ directions. (The pixel centers are placed at coordinates starting at 0 and ending at 12,750 m in each direction). The size of the model in the vertical direction is variable and depends on the number and thicknesses of units in the randomly generated stratigraphic column and on the regional dip.

Each model initially consists of a succession of subhorizontal layers. The number of layers and the thickness of each unit are randomly chosen. A regional dip and dip direction are also chosen. The dip and dip direction are the same for all units.

Thus, the unit tops are a series of parallel planes, and one can easily calculate which two planes any given ($x$,$y$,$z$) point is between and therefore which unit it is in.

The basic fold shape is sinusoidal in both the along-axis and axis-perpendicular directions. This is then further modified by parameters to specify asymmetry and sigmoidal shape. The sigmoidal shape parameters are, however, only used in making the sigmoidal fold example synthetic model and are not used for generating the random models. The fold style is

similar folding with a vertical axial plane, so the deformation consists purely of uplift without horizontal motion and is a function of only $x$ and $y$, not $z$.

To calculate the fold-related uplift, we work in a rotated coordinate system:

$$\begin{aligned} x^{'} &= (x - x_c) \cos(-v) - (y - y_c) \sin(-v) \\ y^{'} &= (x - x_c) \sin(-v) + (y - y_c) \cos(-v) \end{aligned}, \tag{A1}$$

where ($x$,$y$) is a position at which to calculate uplift, ($x_c$,$y_c$) is the location of the center of the fold, and $v$ is the fold vergence

direction as measured counterclockwise from the $x$-axis. (For simplicity, we use this angle rather than the azimuth as measured clockwise from north.)

The size of the fold is defined by two wavelengths: $\lambda_x$, which gives the wavelength in the direction perpendicular to the fold axis, and $\lambda_y$, which gives the wavelength in the direction parallel to the fold axis. If the fold is asymmetric, we then calculate two additional wavelengths that are used for the two sides of the fold in order to make it asymmetric:

$$\lambda_1 = \frac{2\lambda_x}{1+\frac{1}{a}}, \tag{A2}$$

$$\lambda_2 = \frac{2\lambda_x}{1+a}, \tag{A3}$$





where the value of $a$ controls the asymmetry. If the fold is not asymmetric, we set $\lambda_1 = \lambda_2 = \lambda_x$. We also calculate an offset fold axis $x_c'$ coordinate:

$$x_c' = \frac{\lambda_2 - \lambda_x}{2}. \tag{A4}$$

If the fold has a sigmoidal shape, we calculate an $x_c'$ that is a function of $y'$ using the equation:

$$x_c' = \frac{\lambda_2 - \lambda_x}{2} + \frac{A_\sigma}{1 + e^{\frac{k_\sigma y'}{s_\sigma}}} - \frac{A_\sigma}{2}. \tag{A5}$$

In this equation, $A_\sigma$ is an amplitude term, $k_\sigma$ is a steepness parameter, and $s_\sigma$ is a scale parameter for distance in the $y'$ direction. The first term of this equation is the asymmetry offset from above, the second term is a logistic function, which produces the sigmoid shape, and the third term centers the offset due to the logistic function so that it is 0 at $y' = 0$.

535       With the wavelengths and fold axis defined, we calculate uplift ($U_{fold}$) at each point $(x', y')$ as:

$$U_{fold} = \begin{cases} A\left(\frac{1}{2}\right)\left(1 + \cos\left(\frac{2\pi(x' - x_c')}{\lambda_i}\right)\right)\left(\frac{1}{2}\right)\left(1 + \cos\left(\frac{2\pi(y' - y_c')}{\lambda_y}\right)\right), & r \leq 1 \\ 0, & r > 1 \end{cases}, \tag{A6}$$

where $A$ is the fold amplitude and $r$ is defined as

$$r = \sqrt{\left(\frac{x' - x_c'}{\lambda_i}\right)^2 + \left(\frac{y' - y_c'}{\lambda_y}\right)^2}, \tag{A7}$$

and

$\lambda_i = \begin{cases} \lambda_1 & x' > x_c' \\ \lambda_2 & x' \leq x_c' \end{cases}.$ \hfill (A8)

If the model contains multiple folds, then the total uplift is the sum of all their individual uplifts.

          The model for synthetic faults is a simple vertical dip-slip fault with an elliptical displacement field in the along-strike direction, a constant displacement field in the vertical direction, and a volumetric displacement field that decreases away from the fault (Cardozo et al., 2008). It is a simplification of the method used by Wu et al. (2020). We begin by rotating points

from the map coordinates $(x,y)$ to the rotated coordinates $(x',y')$ as in Eqn. A1, where $v$ is now the fault strike direction. We then calculate the distance in the strike-direction from the fault center:

$$r = \left|\frac{x'}{L}\right|, \tag{A9}$$

where $L$ is the fault length.

          We then calculate displacement along the fault ($d$) as:

$d = \begin{cases} 2d_{\max}(1 - r)\sqrt{\frac{(1+r)^2}{4} - r^2}, & r \geq 1 \\ 0, & r < 1 \end{cases}, $ \hfill (A10)





where $d_{max}$ is the maximum displacement. Eqn. A10 is a one-dimensional version of Eqn. 6 of Wu et al. (2020). When randomly choosing $d_{max}$, we calculate it from $L$ as

$$d_{\max} = 10^{\gamma} L, \tag{A11}$$

so the randomly chosen value is $\gamma$, rather than $d_{max}$ directly, and it is chosen to be consistent with real-world fault scaling

(Cowie and Scholz, 1992).

With this, we calculate displacement away from the fault ($D$), following Cardozo et al. (2008) as:

$$D = \begin{cases} d\left(1 - \frac{|y'|}{R}\right)^2, & \frac{|y'|}{R} \le 0 \\ 0, & \frac{|y'|}{R} > 0 \end{cases}, \tag{A12}$$

where $R$ is the radius to which faulting extends. $R$ is randomly chosen as a fraction of $L$.

Finally, we partition $D$ between the two sides of the fault and calculate the total uplift as

$$U_{fault} = \begin{cases} \alpha D, & y' > 0 \\ (\alpha - 1)D & y' \le 0 \end{cases}, \tag{A13}$$

where $\alpha$ controls the fraction of displacement in the hanging wall vs. the footwall (or simply one side vs. the other for these vertical faults).

To avoid over-fitting and to make the model more robust to detecting folds that may not perfectly match our synthetic fold shape, we add noise to the uplift. This noise takes the form:

$$U_{noise} = An_1 + BU_0 n_2, \tag{A14}$$

where $A$ and $B$ are amplitude parameters that are randomly chosen for each model, $n_1$ and $n_2$ are random numbers drawn from a standard normal distribution at each pixel, and $U_0$ is the uplift before adding the noise.

Topography is randomly created using Perlin noise, with a different random seed for each randomly generated map. In addition to the random seed, the noise is controlled by four parameters: the cell size, the number of octaves, the lacunarity,

and the persistence. The cell size (in pixels) defines the size of the grid cells used to generate Perlin noise and controls the frequency of the noise, the number of octaves is the number of successive levels of noise that are generated, the lacunarity controls how the frequency of the noise changes with each octave, and the persistence controls how the amplitude changes with each octave. Assuming that lacunarity is $> 1$ and persistence is $< 1$, successive octaves will have higher frequency but lower amplitude, thus increasing the level of small-scale detail in the topography with an increasing number of octaves.

Since the Perlin noise function generates values between -1 and 1, it is then multiplied by a randomly chosen topographic amplitude. It is also centered so that the mean topography is at 75% of the pre-deformation vertical thickness of the stratigraphic column.

For each model, a random number of folds ($N_{folds}$) and a random number of faults ($N_{faults}$) are chosen along with random values for all their parameters. The total uplift at each ($x,y$) coordinate is the sum of the contributions of all folds (Eqn.

A6), faults (Eqn. A13), and noise (Eqn. A14).





$$U(x,y) = \sum_{i=1}^{N_{folds}} U_{fold,i}(x,y) + \sum_{j=1}^{N_{faults}} U_{fault,j}(x,y) + U_{noise}(x,y). \tag{A16}$$

Since uplift is the only deformation in the models, we can calculate the pre-deformational elevation of any point on the map by subtracting the uplift from the elevation of the topography at that point. With the pre-deformational elevation, we determine which unit it is in based on the original stratigraphy.

Quaternary units, assigned a relative age of 0, are added to the synthetic maps in the training dataset for the convolutional neural network in order to train it to deal with these units, which often cut across geological structure on real-world maps. We randomly simulate these by creating a second random Perlin noise field and assigning a Quaternary unit to all pixels with a value above a certain percentile of the values in this Perlin noise field. The percentile is randomly varied so that some maps have more Quaternary deposits than others.


**Table A1: Random model parameters.** $U\{a,b\}$ indicates a discrete uniform distribution between a and b, inclusive of both a and b. $U[a,b)$ indicates a continuous uniform distribution between a and b, inclusive of a but not b. $N(\mu,\sigma)$ indicates a normal distribution with mean, μ, and standard deviation, σ. $p(i)$ indicates the probability of the discrete value i.

| Parameter | Variable | Distribution |
|---|---|---|
| Number of geological units | | $U\{5, 25\}$ |
| Thickness of each unit (m) | | $U[50, 800)$ |
| Regional dip direction (°) | | $U[0, 360)$ |
| Regional dip (°) | | $N(0, 1)$ |
| Topography amplitude (m) | | $U[100, 2000)$ |
| Uplift noise amplitude (m) | $A$ | $U[0, 100)$ |
| Uplift noise fraction, (m) | $B$ | $U[0, 0.75)$ |
| Number of folds | $N_{folds}$ | $p(0)=0.15, p(1)=0.425, p(2)=0.2975, p(3)=0.1275$ |
| Fold wavelength (m) | $\lambda_x$ | $U[1000, 6000)$ |
| Along-axis fold wavelength (m) | $\lambda_y$ | $U[5000, 20000)$ |
| Fold amplitude | $A$ | $U[200, 250)$ |
| Fold vergence direction (°) | $v$ | $U[0, 360)$ |
| Fold center x coordinate (m) | $x_c$ | $U[50, 12700)$ |
| Fold center y coordinate (m) | $y_c$ | $U[50, 12700)$ |
| Fold asymmetry | $a$ | $N(0,0.3)$ |
| Number of faults | $N_{faults}$ | $p(0)=0.5, p(1)=0.35, p(2)=0.105, p(3)=0.045$ |
| Fault length (m) | $L$ | $U[1500, 20000)$ |
| Fault scaling exponent | $\gamma$ | $U[-2, 0]$ |
| Fault strike (°) | | $U[0, 360]$ |
| $R$ fraction of $L$ | | $U[0.2, 0.5]$ |
| Displacement asymmetry | $\alpha$ | $U[0, 1]$ |
| Perlin noise octaves | | $U[2, 8]$ |
| Perlin noise lacunarity | | $U[1, 3]$ |
| Perlin noise persistence | | $U[0, 1]$ |
| Perlin noise cell size (pixels) | | $U[64, 512]$ |
| Percentile for Quaternary (%) | | $U[0,25]$ |






**Code Availability**

The current version of the GEOMAPLEARN code is available for download on GitLab at https://doi.org/10.18144/8aee-7b77.
The specific version of the code used to produce the results described in this paper is archived on Zenodo at

https://doi.org/10.5281/zenodo.11073379. The code is distributed under a CeCILL – B free software license.

**Data Availability**

The code necessary to generate all the synthetic model data shown in this paper is provided as part of the GEOMAPLEARN code distribution. The Lavelanet and Esternay map shapefiles were provided to us by the BRGM but are not licensed for redistribution. Anyone interested in using those data should email contact@brgm.fr to inquire about access.

**Author contribution**

**David Oakley:** Methodology, Software, Writing – original draft, **Christelle Loiselet:** Conceptualization, Methodology, Funding acquisition, Supervision, Writing – review & editing, **Thierry Coowar:** Methodology, Software, **Vincent Labbe**: Methodology, Software, Writing – review & editing, **Jean-Paul Callot:** Conceptualization, Funding acquisition, Supervision, Writing – review & editing

**Competing interests**

The authors declare that they have no conflict of interest.

**Acknowledgements**

This research is funded by an inter-Carnot grant (contract number: CONT-2021-0089) from the Carnot ISIFoR and BRGM and was supported by E2S UPPA and the Total Structural geology chair at UPPA (JP Callot) and by BRGM funding.



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
