# Peer review of "GEOMAPLEARN 1.0: Detecting geological structures from geological maps with machine learning"

_Geoscientific Model Development, 2024_

## Author Response (AR1)

Reviewer comments are in bold with our responses beneath them.

**Reviewer 1 (Anonymous Referee #1):**

**Major Comments:**

**- The title of the paper refers to the detection of geological structures, whereas the rest of the document focuses only on the detection of fold structures. The authors make clear that an extension to more general settings is desired for the future. It is clear how this can be achieved for the supervised approach, but might prove very challenging for the unsupervised technique. For isolated geological structures, this is likely possible but a combination of different structures might yield to problems in the unsupervised approach and potentially also for the supervised techniques. Therefore, it would be highly advantageous to have an example combining more than one geological feature to be able to judge the potential capabilities of both approaches in a more general setting. Without this example, it is challenging to see whether especially the presented unsupervised approach is extendable to complex structures or limited to more simple settings.**

The extension to other structures is certainly a future goal, but it is beyond the scope of this paper and will require significantly more work. We have modified the title of the paper to refer specifically to folds.

**- For the unsupervised approach, the first step extracts rays. For the application to other studies, it is interesting to know what distance between the rays is in general desired. Are there any rules of thump? How would a too large distance affect the results? And would a too small distance significantly impact the efficiency/cost of the approach?**

We do not have an exact rule, but in general more are better, but at a cost in computational time. We have added some discussion of this issue in the second paragraph of section 2.2.

**- For the supervised approach, it would be good to extend the results/discussion on the aspect whether there is a dependency on the hyperparameters. Furthermore, additional details regarding the architecture of the U-Net should be provided (e.g., number of hidden layers, number of neurons per layer, learning rate, …).**

We have added these details and a discussion of the hyperparameters in section 2.3, and we have added several more plots to Figure 4 to show how the training loss changes as the hyperparameters are changed.

**Minor Comments:**

**- Abstract: Not every reader might be familiar with the hdbscan clustering algorithm. Therefore, it would be useful to add a very brief explanation in the abstract.**

There is not much room in the abstract to describe the algorithm, but we have added a statement of the distinguishing feature that led us to choose this algorithm: the fact that it does not require a number of clusters to be specified a priori, as most other clustering algorithms do.

**- Abstract: The abstract is a bit generic and specified to better highlight the novelty of the approach presented in the paper.**

We have expanded the abstract.

**- Introduction: The authors present previous work for both the automatic detection of geological structures from geological maps and the automatic classification of lithologies from remote sensing and geophysical data. It would be useful to extend this to include also the usage of machine learning approaches in the field of geological modeling. Especially since in this field also unsupervised approaches (Wang et al., 2017) have been tested, and it would be interesting to know if these approaches could potentially work also in the current settings.**

We added a reference to this paper as well as some recent papers that have used neural networks for 3D geological modeling.

**- Equations 1 and 2: It might be better to move the description of equations 1 and 2 above the equations. Otherwise, these equations might be confusing for the reader at first since the type of notation might not be expected.**

The terms are explained immediately below, which seems more typical of how I have usually seen terms in equations defined. We have clarified before the equations that these are regular expressions, so that the reader will not be expecting a mathematical equation.

**- Figure 1A: It would be good to add a verb to "Intersection of grid and map polygons" to unify this point with the rest of the figure.**

We changed this to say "Find intersections of grid and map polygons.

**- Figures:  The resolution of some figures is relatively low. It might be advantageous to switch from bitmaps to vector graphics.**

We have produced vector graphics forms of the final figures, saved as pdf.

**Reviewer 2 (Anonymous Referee #2):**

**Major Comments:**
**1) In the scheme of the grid of sample lines, the recognition efficiency is low when there are many nodes; When there are few nodes, it is easy to miss folds. Folded strata are usually distributed in long strips due to compression (except for some special domes). Therefore, based on the direction of the exposed strata, drawing sample lines is expected to reduce the number of sample lines and improve recognition efficiency. Suggest adding relevant discussions during the discussion;**

It is true that sampling perpendicular to fold axes would be more efficient. However, this requires prior interpretation of at least some folds, and our goal is to develop methods that can detect folds automatically without prior knowledge. Since we have used a large number of sampling lines, we do not expect missing folds to be a major problem, and we have not seen this as an issue in our results. To clarify this, we have added some discussion about the need for a large number of sampling lines in the second paragraph of section 2.2.

**2) The use of 250m as the distance (lines 90-95) in attitude calculation may be appropriate for the experiments in this paper. The value of this parameter should be related to the scale of folds and the scale of the map. Geological maps of different scales reflect fold structures of different scales, and their values should be different. It is recommended to add appropriate discussions;**

We had already stated that this value "is likely to be map dependent." We have expanded on this to specify the scales of the map, the folds, and the topography as factors that may affect it.

**3) Unsupervised learning methods, due to the use of clustering algorithms, limit the automation level of the algorithm. In this scheme of Figure 3, only the midpoints of the possible fold segments were identified. If the boundary points of the folds are identified simultaneously, it is possible to segment the midpoints through the boundary points, which can avoid clustering processing and improve the automation level of the algorithm possibly;**

We do identify boundary points, as we use them to find the midpoints (between two boundary points). We did do some experiments with trying to cluster the boundary points instead of the midpoints, but this did not work very well. The clustering expects single points, rather than pairs of end points, so we found that using the midpoints works best for clustering.

**4) The horizontal rock layers (horizontal structures) distributed along both sides of the valley may produce patterns similar to folds in geological maps. Suggest giving appropriate consideration.**

This is an issue that we already addressed in the paper. Specifically, the "flat model" test maps (Fig. 6A-B) and the Esternay map are used to test this issue. It is discussed in section 4.3.

**Minor Comments:**
**1) The paper mainly explores the automatic detection method of folds, and it is suggested that the keyword of the fold be included in the title of the paper;**

We have modified the title to include the keyword fold.

**2) Why focus on detecting geological folds? Please state the appropriate reasons in the manuscript;**

We added a sentence about this at the end of section 1.

**3) How to combine the two methods? Please provide more suggestions.**

We do not intend to combine the methods, as they work quite differently. Our intended focus is on comparing them and providing two different alternatives.

**Reviewer 3 (Guillaume Caumon):**

**General Comments:**

**- Estimating the dip of layers from unit polygons is a good idea but may be sensitive to noise or degenerate configurations (n aligned points only constrain one of the two orientation angles to describe the surface orientation). It also relies on labels on the (closed) unit polygons to make sure that faults and unconformities are properly handled. Please add a few more details about how this is done (See also Fernandez, Journal of Structural Geology 2005).**

We attempt to avoid degenerate configurations and minimize the effects of noise by sampling points over a sufficiently large distance. It is likely that there are still some poor-quality measurements, but our clustering analysis can exclude outliers, and our identification of fold axes averages over points scattered around the axis location, so a few bad measurements are unlikely to have a substantial effect on the final result. We have added some discussion of these issues and references to the Fernandez paper in section 2.2 where we describe the method.
We have not extensively tested our method's ability to handle faults and unconformities. A few are unlikely to be a major problem, as they won't tend to produce fold-like patterns of oppositely dipping matched contacts on either side of a common axis. However, maps with a large number of faults or unconformities would likely cause problems for our method in its current state. This is an issue for future work, which we discuss in Section 4.4, but is beyond the scope of this fold-focused paper.

**- It could be relevant to include and discuss how the proposed method compares to Jessel et al. (GMD, 2021 - map2loop paper). More fundamentally, geometric 3D interpolation from map data could also automatically generate fold geometry and the associated characteristics without the need for pre-computing surface orientations, see Caumon et al., IEEE Trans. Geosci and Remote Sensing 2013.**

These are certainly related works, and we have added a sentence about them to the introduction. The map2loop software seems to be a method for extracting and interpolating data, while our proposed method is for conceptually interpreting it. The two methods could potentially be complementary, as data extracted from map2loop could be interpreted by our method, and structural interpretation can help to guide interpolation in areas of sparse data. Another interesting possibility could be the use of the scalar field produced by these methods as an input to the convolutional neural network. We added a sentence about this to the end of section 4.4.

**- The scan line strategy proposed in the unsupervised method is nice to generate uniformly oriented lines. I wonder if it generates uniformly located lines, and whether varying line densities could be a source of sampling bias. (A way to check could be to estimate the number of times that each pixel of the map is intersected by a line). Given the large number of lines used in the paper, this is probably not a big problem, but I prefer asking.**

We tried making such a plot. The density of intersections is greatest closest to the edges of the circle. This can even be seen in Figure 1B to some extent. Attempting to sample with an even distribution in space would introduce bias in orientation, so there is no perfect solution that we are aware of. The best solution is probably just to have a large number of lines in total so that all areas of the map get sampled enough times to produce a cluster of the specified minimum size. We have added discussion of this issue in the second paragraph of section 2.2.

**- The training and data assume that no growth strata are present on the anticline. Is there a way to account for unconformities with the proposed regular expression approach? Adding some elements on this would be welcome in the discussion.**

The regular expression approach would not necessarily be hindered by growth strata or unconformities, although we have not tested those situations. It works by identifying matched contacts, meaning that two adjacent units must occur on either side of the fold. This could include the contact of growth with pre-growth strata on either side of the anticline, if the pre-growth unit is the same on both sides. Even if an unconformable contact is not properly identified as part of a fold, the fold will still be identified if there are a sufficient number of folded conformable contacts within the sedimentary sequence. We had already mentioned unconformities in section 4.4 as one of several issues not addressed by the current method, and we have added growth strata to this list as well.

**- Instead of constraining the dip of layers in the unsupervised method, could you consider the difference between the topographic and layer slopes along the scan line? Indeed, this difference is what makes the distinction between a true fold and an apparent fold.**

While this might help in some situations, our existing method has worked sufficiently well for the examples tested. It is also likely that we would still need to constrain at least the dip directions, as we compare these across the two sides of the fold to reject anticlines or synclines without the expected dip directions.

**- If I understand correctly, the '?' In the regular expression may possibly lead to underestimating the lateral extent of the anticlines / sync lines in the unsupervised method when more than 4 units are folded. However, this does not seem to happen. Please comment.**

The lateral extent of the anticline or syncline is based on the convex hull of the matched contacts. (See the last sentence of section 2.2.) Any individual match may, indeed, not extend across the entire width of the fold. However as long as the outermost folded contact is included in the analysis, then the convex hull of all folded contacts will cover the full area affected by folding. Where a single ray crosses multiple folded contacts, all of them will be identified: not just the innermost or outermost one. We added a sentence about this in section 2.2, saying: "If there are multiple contacts on either side of the same fold, there will be multiple overlapping segments, with the outermost one crossing the full width of the fold."

The main purpose of the '?' is to avoid matching limbs of two different folds in a train of parallel folds.

**- In several figures, only the geology is shown, which makes the 3D perception of the topgraphy difficult. If possible, it would be nice to overlay topographic contours onto the colored geological units (as in classical geological maps) to help the reader better perceive the various configurations.**

We have added this to Figure 2C. However, in many of the other figures, with many of the maps being rather small, the contours were either hard to see or obscured the maps too much. We have, instead, added separate subfigures showing the elevation in Figures 6, 11, and 14.

**- I like the discussion between the relative merits of the supervised and unsupervised methods. To echo a point of the discussion, I hypothesize that the independence assumption between the geological model and the topography could make « realistic » intersection patterns (as determined by erosion) marginal in the training set. If so, I suspect that generating more realistic erosion patterns and the associated geomorphological features in the training set could possibly help the CNN. Another potential bias may come from the topography which is always located at 75% of the rock thickness. Maybe making this average elevation random could be an computationally efficient way to augment the training data set and generate more representative configurations. As these suggestions may or may not improve the results, I don't see their implementation as needed for the paper to be publised.**

This is a good point. The Perlin-noise-based synthetic topography is easy to generate rapidly, but it is only a rough approximation of real geomorphology. More realistic synthetic topography in the training dataset would probably improve the ability of the CNN to deal with geomorphologically complex real-world maps, such as the Esternay example. We have added a comment to this effect at the end of section 4.3.
The location of the topography at 75% of the rock thickness is unlikely to be a major issue because the number and thickness of stratigraphic units is randomly generated for each model. 75% was chosen so that both the highest elevations of the topography and the deepest units exposed in fold cores would generally be within the range of the modeled stratigraphy.

**- Lines 380-385: Could simply turning the training data upside down be done to detect synclines ?**

We did experiment with this briefly, but we had some difficulties due to synclines being more often formed as the space between anticlines than by actual downward folding. We also could not simultaneously identify anticlines and synclines with the CNN classification method we used, since they overlap and only one class can be assigned per pixel. (The limb of an anticline is also the limb of an adjacent syncline.) We discuss this issue briefly in section 2.3 (fourth paragraph). However, it is certainly an issue deserving of more work in the future.

**- Line 419: small typo: 'Table S1' should read Table A1.**

We have fixed this mistake.

**- I agree with some previous comments that call for choosing a more specific title.**

We have modified the title to refer to folds specifically.